# Description of Residual Stress and Strain Fields in FGM Hollow Disc Subject to External Pressure

**DOI:** 10.3390/ma12030440

**Published:** 2019-01-31

**Authors:** Stanislav Strashnov, Sergei Alexandrov, Lihui Lang

**Affiliations:** 1Department of Civil Engineering, Peoples’ Friendship University of Russia (RUDN University), Miklukho-Maklaya st. 6, 117198 Moscow, Russia; 2School of Mechanical Engineering and Automation, Beihang University, No. 37 Xueyuan Road, Beijing 100191, China; sergei_alexandrov@spartak.ru (S.A.); lang@buaa.edu.cn (L.L.); 3Ishlinsky Institute for Problems in Mechanics, 101-1 Prospect Vernadskogo, 119526 Moscow, Russia

**Keywords:** hollow disc, external pressure, residual stress, residual strain, flow theory of plasticity

## Abstract

Elastic/plastic stress and strain fields are obtained in a functionally graded annular disc of constant thickness subject to external pressure, followed by unloading. The elastic modulus and tensile yield stress of the disc are assumed to vary along the radius whereas the Poisson’s ratio is kept constant. The flow theory of plasticity is employed. However, it is shown that the equations of the associated flow rule, which are originally written in terms of plastic strain rate, can be integrated with respect to the time giving the corresponding equations in terms of plastic strain. This feature of the solution significantly facilitates the solution. The general solution is given for arbitrary variations of the elastic modulus and tensile yield stress along the radial coordinate. However, it is assumed that plastic yielding is initiated at the inner radius of the disc and that no other plastic region appears in the course of deformation. The solution in the plastic region at loading reduces to two ordinary differential equations. These equations are solved one by one. Unloading is assumed to be purely elastic. This assumption should be verified a posteriori. An illustrative example demonstrates the effect of the variation of the elastic modulus and tensile yield stress along the radius on the distribution of stresses and strains at the end of loading and after unloading. In this case, it is assumed that the material properties vary according to power-law functions.

## 1. Introduction

Stress and strain analyses of solid and hollow circular discs have long been an important topic in the mechanics of solids. The motivation of doing such analyses is that circular discs subject to mechanical, thermal, and inertial loading are used in many sectors of industry. The performance of such discs under service conditions can be improved by using functionally graded materials. The material may be continuously graded or be piecewise homogeneous. It is assumed in the present paper that the distribution of all material properties is axisymmetric. Discs made of homogeneous materials are not discussed. Discs made of functionally graded materials have been the subject of intense research. A linearly elastic solution under plane stress and plane strain conditions has been given in [1], assuming that the disc is loaded by external or internal pressure. It has been concluded that the stress response of the functionally graded disc is significantly different from that of the homogeneous disc. Another plane stress solution of this boundary value problem has been obtained in [2] and another plane strain solution in [3]. A thermoelastic stress solution for a disc of variable thickness has been found in [2]. A thermoelastic analysis of a disc subject to a steady-state temperature distribution together with external and internal pressures has been provided in [4]. It has been assumed in this work that the material properties are arbitrary smooth functions of the radial coordinate. A similar boundary value problem for a multilayered hollow cylinder has been solved in [5]. A design driven by the minimization of induced stresses in elastic multilayer cylinders under plane stress conditions has been proposed in [6]. All of the solutions above are purely elastic or thermoelastic. The process of autofrettage of a functionally graded cylinder has been studied in [7]. The analysis of this process requires the use of an elastic/plastic model. In [7], the deformation theory of plasticity together with the von Mises yield criterion has been employed. Another elastic/plastic plane strain solution for a functionally graded cylinder has been given in [8]. The solution is based on Tresca’s yield criterion, which significantly simplifies the analysis even in the case of the flow theory of plasticity. 

There is a vast amount of literature on functionally graded rotating discs. The elastic response of an arbitrary functionally graded polar orthotropic disc has been investigated in [9]. Another purely elastic solution has been given in [10], using the finite difference method. Thermoelastic analyses have been presented in [11,12,13,14]. The effect of a non-uniform heat source on thermoelastic behavior of a functionally graded rotating disc has been investigated in [15]. The effect of viscosity on the response of a functionally graded rotating disc of variable thickness has been studied in [16]. The limit of elastic angular velocity has been determined in [17]. The effect of variable angular velocity on the elastic response of a functionally graded rotating disc has been analyzed in [18]. A design driven by weight optimization of a disc subject to thermomechanical loading has been proposed in [19]. Most of the available elastic/plastic solutions fall into three categories. A series of solutions is devoted to discs obeying Tresca’s yield criteria [20,21]. As it has been mentioned before, the use of Tresca’s yield criterion significantly simplifies the solution. Another category includes the solutions for the deformation theory of plasticity [22,23,24,25]. In some cases, using deformation theories of plasticity is justified since the stress path is nearly proportional. However, it has been shown in [26] that it may not be so in thin discs. The third category includes stress solutions [27,28]. In this case, no flow rule is necessary to find the solution.

An advantage of the present elastic/plastic solution is that the flow theory of plasticity in conjunction with the von Mises yield criterion is employed. It is assumed that a hollow disc is subject to external pressure, followed by unloading. First, the general solution is derived under plane stress conditions assuming that the elastic modulus and tensile yield stress are arbitrary smooth functions of the radial coordinate. It is, however, assumed that plastic yielding initiates at the inner radius of the disc and that there is one plastic region throughout the process of deformation. The Poisson’s ratio is supposed to be constant. This is a typical assumption for functionally graded discs [1,3,10,19]. Second, a numerical example is given assuming that the material properties vary according to power-law functions. This is also a typical assumption for functionally graded discs [1,9,10,11,12,13]. The solution found can be considered as an extension of the solution provided in [1] to the plastic range.

## 2. Statement of the Problem 

Consider a thin hollow disc of functionally graded material subject to uniform pressure $p_{0}$ over its outer radius $b_{0}$, followed by unloading. The inner radius of the disc is denoted as $a_{0}$. The thickness of the disc is constant. The mechanical properties of the disc are classified in terms of the yield stress tension $\sigma_{Y}$, Poisson’s ratio ν, and Young’s modulus *E.* It is assumed that the Poisson’s ratio is constant, whereas the value of both $\sigma_{Y}$ and *E* vary with radius. It is convenient to use a cylindrical coordinate system (r, θ, z) whose z-axis coincides with the axis of symmetry of the disc. The normal stresses in this coordinate system are the principal stresses. The state of stress is plane (i.e., the axial stress in the cylindrical coordinate system vanishes). Therefore, Hooke’s law can be written as
(1)$$\varepsilon_{r}^{e}=\frac{\sigma_{r}-\nu \sigma_{\theta }}{E},\;\varepsilon_{\theta }^{e}=\frac{\sigma_{\theta }-\nu \sigma_{r}}{E},\;\varepsilon_{z}^{e}=-\frac{\nu \left(\sigma_{r}+\sigma_{\theta }\right)}{E}$$
here $\sigma_{r}$ is the radial stress, $\sigma_{\theta }$ is the circumferential stress, $\varepsilon_{r}^{e}$, $\varepsilon_{\theta }^{e}$, and $\varepsilon_{z}^{e}$ are the elastic strains referred to the cylindrical coordinate system. Plastic yielding is controlled by the von Mises yield criterion. Under a plane stress condition, this criterion reads
(2)$$\sigma_{r}^{2}+\sigma_{\theta }^{2}-\sigma_{r}\sigma_{\theta }=\sigma_{Y}^{2}$$

The flow theory of plasticity is adopted. The flow rule associated with the yield criterion (2) is
(3)$$\xi_{r}^{p}=\lambda \left(2\sigma_{r}-\sigma_{\theta }\right),\;\xi_{\theta }^{p}=\lambda \left(2\sigma_{\theta }-\sigma_{r}\right),\;\xi_{z}^{p}=-\lambda \left(\sigma_{r}+\sigma_{\theta }\right)$$
here $\xi_{r}^{p}$, $\xi_{\theta }^{p}$, and $\xi_{z}^{p}$ are the plastic strain rates referred to the cylindrical coordinate system and λ is a non-negative multiplier. The total strain components in the cylindrical coordinate system are given by
(4)$$\varepsilon_{r}=\varepsilon_{r}^{e}+\varepsilon_{r}^{p},\;\varepsilon_{\theta }=\varepsilon_{\theta }^{e}+\varepsilon_{\theta }^{p},\;\varepsilon_{z}=\varepsilon_{z}^{e}+\varepsilon_{z}^{p}$$
here $\varepsilon_{r}^{p}$, $\varepsilon_{\theta }^{p}$, and $\varepsilon_{z}^{p}$ are the plastic strains referred to the cylindrical coordinate system. The constitutive equations should be complemented with the equilibrium equation
(5)$$\frac{\partial \sigma_{r}}{\partial r}+\frac{\sigma_{r}-\sigma_{\theta }}{r}=0$$
and the equation of strain compatibility of the form
(6)$$r\frac{\partial \varepsilon_{\theta }}{\partial r}=\varepsilon_{r}-\varepsilon_{\theta }$$

The boundary conditions at the stage of loading are
(7)$$\sigma_{r}=-p_{0}$$
for $r=b_{0}$ and
(8)$$\sigma_{r}=0$$
for $r=a_{0}$. The boundary conditions at the stage of unloading will be formulated in Section 6. 

It is convenient to introduce the following dimensionless quantities:(9)$$\rho =\frac{r}{b_{0}},\;a=\frac{a_{0}}{b_{0}},\;k=\frac{\sigma_{0}}{E_{0}},\;p=\frac{p_{0}}{\sigma_{0}}$$
here $\sigma_{0}$ is the value of $\sigma_{Y}$ at $r=b_{0}$ and $E_{0}$ is the value of *E* at $r=b_{0}$. Then, the variation of $\sigma_{Y}$ and *E* with ρ can be represented as
(10)$$\sigma_{Y}=\sigma_{0}\Phi \left(\rho \right)\;\mathrm{and}\;E=E_{0}\eta \left(\rho \right)$$
where Φ(ρ) and η(ρ) are arbitrary functions of ρ satisfying the conditions Φ(ρ)=1 and η(ρ)=1 at ρ=1. In what follows, it is assumed that these functions are such that plastic yielding initiates at the inner radius of the disc and no other plastic region appears in continued deformation. 

## 3. Purely Elastic Solution

When *p* is small enough, the entire disc is elastic. In this case, the total strains are equal to the elastic strains. The system of equations comprises Hooke’ law, the equilibrium equation, and the equation of strain compatibility. Using (9) it is possible to rewrite Equations (5) and (6) as
(11)$$\frac{\partial \sigma_{r}}{\partial \rho }+\frac{\sigma_{r}-\sigma_{\theta }}{\rho }=0$$
and
(12)$$\rho \frac{\partial \varepsilon_{\theta }^{e}}{\partial \rho }=\varepsilon_{r}^{e}-\varepsilon_{\theta }^{e}$$
Eliminating the strains in this equation by means of (1) and using (9) results in
(13)$$\frac{\partial \sigma_{\theta }}{\partial \rho }+\frac{\nu }{\rho }\left(\sigma_{r}-\sigma_{\theta }\right)+\frac{\eta }{\rho }\left(\sigma_{\theta }-\nu \sigma_{r}\right)\frac{\partial \left(\rho /\eta \right)}{\partial \rho }-\frac{\left(\sigma_{r}-\nu \sigma_{\theta }\right)}{\rho }=0$$

Equations (11) and (13) comprise the system for determining the distribution of stresses in the purely elastic disc. Then, the distribution of strains can be readily found from (1) and (9). However, the purely elastic solution is not of interest in the case under consideration. Therefore, the solution to Equations (11) and (13) is only necessary to determine the value of *p* at which plastic yielding is initiated. This value of *p* is denoted as $p_{e}$. By assumption, plastic yielding is initiated at ρ=a. It follows from the boundary condition (8) and the yield criterion (2) that $\sigma_{\theta }=-\sigma_{Y}$ at ρ=a at the initiation of plastic yielding. Using (10), this condition can be rewritten as $\sigma_{\theta }=-\sigma_{0}\Phi \left(a\right)$ at ρ=a. This is one of the boundary conditions of the boundary value problem to be solved. The other boundary condition is given by (8). Equations (11) and (13) should be solved together with these boundary conditions. The value of $p_{e}$ is readily found from this solution as $p_{e}=-\sigma_{r}\left(1\right)/\sigma_{0}$. In what follows, it is assumed that $p>p_{e}$.

## 4. Elastic/Plastic Stress Solution

If $p>p_{e}$, then the disc consists of two regions, elastic and plastic. The elastic region occupies the domain $\rho_{c}\leq \rho \leq 1$ and the plastic region the domain $a\leq \rho \leq \rho_{c}$. Here, $\rho_{c}$ is the elastic/plastic boundary. Consider the plastic region. The yield criterion (2) is satisfied by the following substitution:(14)$$\frac{\sigma_{r}}{\sigma_{0}}=\frac{2\Phi \left(\rho \right)\sin\psi }{\sqrt{3}}\;\mathrm{and}\;\frac{\sigma_{\theta }}{\sigma_{0}}=\Phi \left(\rho \right)\left(\frac{\sin\psi }{\sqrt{3}}+\cos\psi \right)$$
here ψ is a new function of ρ. This function should be found from the solution. Substituting (14) into (11) yields
(15)$$\frac{d\psi }{d\rho }+\frac{\tan\psi }{\Phi }\frac{d\Phi }{d\rho }+\frac{\left(\tan\psi -\sqrt{3}\right)}{2\rho }=0$$
The boundary condition to this equation follows from (8) and (14). In particular, $\sigma_{r}=0$ if ψ=0 or ψ=π. It is evident that $\sigma_{\theta }<0$ at ρ=a. Then, the boundary condition to Equation (15) is
(16)ψ=π
for ρ=a. Solving Equation (15) together with this boundary condition supplies the variation of ψ with ρ. This solution and (14) determine the distribution of the stresses in the plastic region. Let $p_{p}$ be the value of *p* at which the entire disc becomes plastic. Putting in the solution for the radial stress ρ=1 gives the value of $p_{p}$ as $p_{p}=-\sigma_{r}\left(1\right)/\sigma_{0}$. In what follows, it is assumed that $p<p_{p}$. In this case, $a<\rho_{c}<1$. The value of $\sigma_{r}$ on the plastic side of the elastic/plastic boundary is denoted as $\sigma_{r}^{c}$ and the value of $\sigma_{\theta }$ on the plastic side of the elastic/plastic boundary as $\sigma_{\theta }^{c}$. These values are readily found from the solution of (15) and (14). Equations (11) and (13) are valid in the elastic region. The radial and circumferential stresses must be continuous across the elastic/plastic boundary. Therefore, the boundary conditions to Equations (11) and (13) are
(17)$$\sigma_{r}=\sigma_{r}^{c}\;\mathrm{and}\;\sigma_{\theta }=\sigma_{\theta }^{c}$$
for $\rho =\rho_{c}$. Solving Equations (11) and (13) together with these boundary conditions supplies the distribution of the radial and circumferential stresses in the elastic region. In particular, the value of *p* involved in (7) is determined from the equation $p=-\sigma_{r}\left(1\right)/\sigma_{0}$. Therefore, the solution found connects p and $\rho_{c}$. One of these parameters should be prescribed. Then, the other parameter is determined from the solution.

## 5. Elastic/Plastic Strain Solution

Consider the plastic region, $a\leq \rho \leq \rho_{c}$. Eliminating λ between the equations in (3) gives
(18)$$\frac{\xi_{r}^{p}}{\xi_{\theta }^{p}}=\frac{\left(2\sigma_{r}-\sigma_{\theta }\right)}{\left(2\sigma_{\theta }-\sigma_{r}\right)},\;\frac{\xi_{z}^{p}}{\xi_{\theta }^{p}}=-\frac{\left(\sigma_{r}+\sigma_{\theta }\right)}{\left(2\sigma_{\theta }-\sigma_{r}\right)}$$

Using (14), the stresses in these equations can be expressed in terms of ψ. Then, taking into account that $\xi_{r}^{p}=\partial \varepsilon_{r}^{p}/\partial t$, $\xi_{\theta }^{p}=\partial \varepsilon_{\theta }^{p}/\partial t$, and $\xi_{z}^{p}=\partial \varepsilon_{z}^{p}/\partial t$ Equation (18) is transformed to
(19)$$\frac{\partial \varepsilon_{r}^{p}}{\partial t}=\frac{\left(\sqrt{3}\tan\psi -1\right)}{2}\frac{\partial \varepsilon_{\theta }^{p}}{\partial t},\;\frac{\partial \varepsilon_{z}^{p}}{\partial t}=-\frac{\left(\sqrt{3}\tan\psi +1\right)}{2}\frac{\partial \varepsilon_{\theta }^{p}}{\partial t}$$
here *t* is the time. It is seen from the structure of Equation (15) and the boundary condition (16) that ψ is independent of *t*. Therefore, the coefficients of $\partial \varepsilon_{\theta }^{p}/\partial t$ in (19) are also independent of *t*, and the equations in (19) can be immediately integrated with respect to the time to give
(20)$$\varepsilon_{r}^{p}=\frac{\left(\sqrt{3}\tan\psi -1\right)}{2}\varepsilon_{\theta }^{p},\;\varepsilon_{z}^{p}=-\frac{\left(\sqrt{3}\tan\psi +1\right)}{2}\varepsilon_{\theta }^{p}$$

It has been taken into account here that $\varepsilon_{r}^{p}=\varepsilon_{\theta }^{p}=\varepsilon_{z}^{p}=0$ at the elastic/plastic boundary. The elastic strains in the plastic region, $\varepsilon_{r}^{ep}$, $\varepsilon_{\theta }^{ep}$ and $\varepsilon_{z}^{ep}$, are determined from (1) and (14) with the use of (9) and (10). As a result,
(21)$$\begin{array}{l} \varepsilon_{r}^{ep}=k\Lambda \left[\frac{\left(2-\nu \right)}{\sqrt{3}}\sin\psi -\nu \cos\psi \right],\;\varepsilon_{\theta }^{ep}=k\Lambda \left[\frac{\left(1-2\nu \right)}{\sqrt{3}}\sin\psi +\cos\psi \right],\;\\ \varepsilon_{z}^{ep}=-\nu k\Lambda \left(\sqrt{3}\sin\psi +\cos\psi \right) \end{array}$$
here Λ=Φ/η. The total strains are found from (4), (20), and (21) as
(22)$$\begin{array}{l} \varepsilon_{r}=k\Lambda \left[\frac{\left(2-\nu \right)}{\sqrt{3}}\sin\psi -\nu \cos\psi \right]+\frac{\left(\sqrt{3}\tan\psi -1\right)}{2}\varepsilon_{\theta }^{p},\\ \varepsilon_{\theta }=k\Lambda \left[\frac{\left(1-2\nu \right)}{\sqrt{3}}\sin\psi +\cos\psi \right]+\varepsilon_{\theta }^{p},\\ \varepsilon_{z}=-\nu k\Lambda \left(\sqrt{3}\sin\psi +\cos\psi \right)-\frac{\left(\sqrt{3}\tan\psi +1\right)}{2}\varepsilon_{\theta }^{p} \end{array}$$

It follows from these equations that
$$\begin{array}{l} \varepsilon_{r}-\varepsilon_{\theta }=\left(\tan\psi -\sqrt{3}\right)\left[\frac{k\left(1+\nu \right)\cos\psi }{\sqrt{3}}\Lambda +\frac{\sqrt{3}}{2}\varepsilon_{\theta }^{p}\right],\\ \frac{\partial \varepsilon_{\theta }}{\partial \rho }=k\frac{d\Lambda }{d\rho }\left[\frac{\left(1-2\nu \right)}{\sqrt{3}}\sin\psi +\cos\psi \right]+k\Lambda \left[\frac{\left(1-2\nu \right)}{\sqrt{3}}\cos\psi -\sin\psi \right]\frac{d\psi }{d\rho }+\frac{\partial \varepsilon_{\theta }^{p}}{\partial \rho }. \end{array}$$.

These equations and Equation (6), in which *r* should be replaced with ρ by means of (9), combine to give
$$\begin{array}{l} \frac{\partial \varepsilon_{\theta }^{p}}{\partial \rho }-\frac{\sqrt{3}}{2}\frac{\left(\tan\psi -\sqrt{3}\right)}{\rho }\varepsilon_{\theta }^{p}+k\frac{d\Lambda }{d\rho }\left[\frac{\left(1-2\nu \right)}{\sqrt{3}}\sin\psi +\cos\psi \right]+\\ k\Lambda \left[\frac{\left(1-2\nu \right)}{\sqrt{3}}\cos\psi -\sin\psi \right]\frac{d\psi }{d\rho }-\frac{k\left(1+\nu \right)\Lambda \left(\sin\psi -\sqrt{3}\cos\psi \right)}{\sqrt{3}\rho }=0. \end{array}$$

The derivative dψ/dρ in this equation can be eliminated by means of (15). Then,
(23)$$\begin{array}{l} \frac{\partial \varepsilon_{\theta }^{p}}{\partial \rho }-\frac{\sqrt{3}}{2}\frac{\left(\tan\psi -\sqrt{3}\right)}{\rho }\varepsilon_{\theta }^{p}+k\frac{d\Lambda }{d\rho }\left[\frac{\left(1-2\nu \right)}{\sqrt{3}}\sin\psi +\cos\psi \right]-\\ k\Lambda \left[\frac{\left(1-2\nu \right)}{\sqrt{3}}\cos\psi -\sin\psi \right]\left[\frac{\tan\psi }{\Phi }\frac{d\Phi }{d\rho }+\frac{\left(\tan\psi -\sqrt{3}\right)}{2\rho }\right]-\\ \frac{k\left(1+\nu \right)\Lambda \left(\sin\psi -\sqrt{3}\cos\psi \right)}{\sqrt{3}\rho }=0 \end{array}$$

Since ψ has already been determined as a function of ρ in Section 4, (23) is a linear differential equation for $\varepsilon_{\theta }^{p}$. The boundary condition to this equation is
(24)$$\varepsilon_{\theta }^{p}=0$$
for $\rho =\rho_{c}$. Once Equation (23) has been solved, the distribution of $\varepsilon_{r}^{p}$ and $\varepsilon_{z}^{p}$ in the plastic region is found from (20) and the distribution of the total strains from (22).

The distribution of strains in the elastic region is determined from the solution for stress found in Section 4 and Hooke’s law.

## 6. Unloading

Let $p_{f}$ be the value of *p* at the end of loading. Then, the boundary conditions for the stage of unloading are
(25)$$\Delta \sigma_{r}=0$$
for ρ=a and
(26)$$\Delta \sigma_{r}=\sigma_{0}p_{f}$$
for ρ=1. Here, $\Delta \sigma_{r}$ is the increment of the radial stress after unloading ($\Delta \sigma_{\theta }$ will stand for the increment of the circumferential stress). It is assumed that unloading is purely elastic. This assumption should be verified a posteriori. Equations (11) and (13), in which $\sigma_{r}$ should be replaced with $\Delta \sigma_{r}$ and $\sigma_{\theta }$ with $\Delta \sigma_{\theta }$, are valid. An iterative procedure should be used for solving this system of equations together with the boundary conditions (25) and (26). Once this boundary value problem has been solved, the increment of strains is determined from Hooke’s law as
(27)$$\Delta \varepsilon_{r}=\frac{\Delta \sigma_{r}-\nu \Delta \sigma_{\theta }}{E},\;\Delta \varepsilon_{\theta }=\frac{\Delta \sigma_{\theta }-\nu \Delta \sigma_{r}}{E},\;\Delta \varepsilon_{z}=-\frac{\nu \left(\Delta \sigma_{r}+\Delta \sigma_{\theta }\right)}{E}$$

The distribution of residual stresses, $\sigma_{r}^{res}$ and $\sigma_{\theta }^{res}$, is given by
(28)$$\sigma_{r}^{res}=\sigma_{r}^{\left(f\right)}+\Delta \sigma_{r},\;\sigma_{\theta }^{res}=\sigma_{\theta }^{\left(f\right)}+\Delta \sigma_{\theta }$$

Here, $\sigma_{r}^{\left(f\right)}$ is the distribution of the radial stress and $\sigma_{\theta }^{\left(f\right)}$ is the distribution of the circumferential stress at the end of loading. These distributions have been found in Section 4. Substituting (28) into (2) provides the condition to verify that the process of unloading is purely elastic in the form
(29)$${\left(\sigma_{r}^{res}\right)}^{2}+{\left(\sigma_{\theta }^{res}\right)}^{2}-\sigma_{r}^{res}\sigma_{\theta }^{res}-\sigma_{0}^{2}\Phi^{2}\leq 0$$

Here, Equation (10) has been taken into account. 

The distribution of residual strains, $\varepsilon_{r}^{res}$, $\varepsilon_{\theta }^{res}$ and $\varepsilon_{z}^{res}$, is given by
(30)$$\varepsilon_{r}^{res}=\varepsilon_{r}^{\left(f\right)}+\Delta \varepsilon_{r},\;\varepsilon_{\theta }^{res}=\varepsilon_{\theta }^{\left(f\right)}+\Delta \varepsilon_{\theta },\;\varepsilon_{z}^{res}=\varepsilon_{z}^{\left(f\right)}+\Delta \varepsilon_{z}$$

Here, $\varepsilon_{r}^{\left(f\right)}$ is the distribution of the radial strain, $\varepsilon_{\theta }^{\left(f\right)}$ is the distribution of the circumferential strain and $\varepsilon_{z}^{\left(f\right)}$ is the distribution of the axial strain at the end of loading. These distributions have been found in Section 5. 

## 7. Illustrative Example

It is often assumed that material properties vary according to a power law along the radius of the disc [1,9,10,11,12,13]. In the case under consideration one possible variant of this law reads
(31)$$\Phi \left(\rho \right)=\rho^{m}\;\mathrm{and}\;\eta \left(\rho \right)=\rho^{n}$$

In all calculations, ν=0.3, a=0.3, and n=0.3. The value of *m* varies in the range 0≤m≤0.3. It is worthy of note that there is no need to prescribe the values of $\sigma_{0}$ and $E_{0}$ for numerical analysis since the stress components are proportional to $\sigma_{0}$, and the strain components are proportional to *k*. It is assumed that $\rho_{c}=0.8$. Then, the value of $p_{f}$ has been found from the stress solution given in Section 4. The dependence of $p_{f}$ on *m* is presented in Table 1. The variation of the radial and circumferential stresses with ρ at $p=p_{f}$ is depicted in Figure 1 and Figure 2, respectively. It has been verified that the yield criterion (2) is not violated in the elastic region. The variation of the radial, circumferential, and axial strains with ρ at $p=p_{f}$ is depicted in Figure 3, Figure 4 and Figure 5, respectively. 

Using the values of $p_{f}$ found (Table 1) the system of Equations (11) and (13) together with the boundary conditions (25) and (26) has been solved for $\Delta \sigma_{r}/\sigma_{0}$ and $\Delta \sigma_{\theta }/\sigma_{0}$. Having this solution and the radial distribution of the radial and circumferential stresses at the end of loading (Figure 1 and Figure 2) the radial distribution of the residual stresses is determined from (28). The variation of the residual radial and circumferential stresses with ρ is shown in Figure 6 and Figure 7, respectively. Then, it has been verified that the inequality (29) is satisfied in the range a≤ρ≤1. Having the solution for $\Delta \sigma_{r}/\sigma_{0}$ and $\Delta \sigma_{\theta }/\sigma_{0}$, the increment of the strains is determined from (27). It is evident from (10) and (27) that these increments are proportional to *k* introduced in (9). The radial distribution of the residual strains is determined from (30). The variation of the residual radial, circumferential, and axial strains with ρ is shown in Figure 8, Figure 9 and Figure 10, respectively. 

## 8. Discussion

The paper presents a general solution for the distribution of stress and strain in an functionally graded hollow disc subject to external pressure, followed by unloading. The solution is valid for any variation of the yield stress and Young’s modulus with radius if the plastic region initiates at, and then propagates from, the inner radius of the disc. The purely elastic solution is valid if $p\leq p_{e}$. This solution is known, and therefore is not considered in the present paper. However, the general elastic solution is used in the elastic region, $\rho_{c}\leq \rho \leq 1$. The stress solution follows from the solution of Equation (13). Then, the distribution of strains is determined from Hooke’s law shown in (1).

The constitutive equations of the classical flow theory of plasticity are adopted. In particular, Hooke’s law shown in (1) is used to connect the stress components and the elastic strain components. This law is valid in the elastic region. The von Mises yield criterion (2) is adopted in the plastic region. In this region, the stress components are connected to the strain rate components rather than to the strain components. The corresponding constitutive equation is the associated flow rule (3). The total strain components in the plastic region are given by (4). A detailed description of this material model can be found in any textbook on plasticity theory (for example, [29,30]).

For any given functions Φ(ρ) and η(ρ) involved in (10), the distribution of stress and strain in the plastic region, $a\leq \rho \leq \rho_{c}$, can be calculated as follows. Two ordinary differential equations, Equations (15) and (23), should be solved numerically. These equations can be solved one by one. In particular, the dependence of ψ on ρ is found from (15). Then, this numerical function is substituted into (23). As a result, a linear differential equation for the circumferential plastic strain, $\varepsilon_{\theta }^{p}$, is obtained. The solution of Equation (15) supplies the distribution of stresses in the plastic region according to (14). A remarkable feature of the strain solution is that the equations in (18), which are derived from the associated flow rule written in terms of plastic strain rates, can be immediately integrated with respect to the time to result in the equations in terms of plastic strains (Equation (20)). This feature of the solution significantly facilitates the solution. Another remarkable feature of the strain solution is that all strain components are proportional to *k* introduced in (9). It is seen from Equations (22) and (23). Therefore, simple scaling of any strain solution provides the strain solution for similar discs of material with the same other properties and geometry, but any value of *k*. To calculate the distribution of strains, it is first necessary to solve Equation (23) for the circumferential plastic strain. Then, the radial and axial plastic strains are readily found from (20) and the solution of (15). The elastic portions of strain in the plastic region are determined from (14) and Hooke’s law shown in (1). Finally, the total strains follow from (4).

The general solution found is used to find a numerical solution, assuming that the power laws shown in (31) are valid. The effect on *m*-value on the radial and circumferential stresses in an a=0.3 disc at ν=0.3, n=0.3, and $\rho_{c}=0.8$ is illustrated in Figure 1 and Figure 2, respectively. The corresponding values of the external pressure are depicted in Table 1. It is seen from these figures that this effect is quite significant, especially on the circumferential stress. The associated distribution of strains is illustrated in Figure 3, Figure 4 and Figure 5 (the radial strain is depicted in Figure 3, the circumferential strain in Figure 4 and the axial strain in Figure 5). The effect of *m*-value on these distributions is not so pronounced as compared to the stress distributions. This is associated with the imposed condition that $\rho_{c}=0.8$ in all cases. If the value of $p_{f}$ were fixed, then the effect of *m*-value on the strain distributions would be more significant as compared to its effect on the stress distribution. In particular, it is seen from Table 1 that the value of $p_{f}$ is quite sensitive to the value of m.

Residual stress and strain fields after purely elastic unloading are also obtained. These distributions are given by Equations (28) and (30). The validity of the pure elastic solution at unloading should be verified by means of Equation (29). The distribution of residual radial and circumferential stresses is illustrated in Figure 6 and Figure 7, respectively. As in the case of the stress distributions at the end of loading, the effect of *m*-value is most significant on the circumferential stress. The effect of this value on the distribution of the residual radial strain is small (Figure 8). It is moderate in the case of the residual circumferential strain (Figure 9) and large in the case of the residual axial strain (Figure 10).

## 9. Conclusions

Stress and strain fields in an elastic/plastic functionally graded annular disc of constant thickness subject to external pressure are obtained under plane stress conditions. Residual stress and strain fields after purely elastic unloading are also obtained. From this work, the following conclusions can be drawn.A remarkable feature of the strain solution is that the equations in (18), which are derived from the associated flow rule written in terms of plastic strain rates, can be immediately integrated with respect to the time to result in the equations in terms of plastic strains (Equation (20)). This significantly facilitates the solution.Another remarkable feature of the strain solution is that all strain components are proportional to *k* introduced in (9). Therefore, simple scaling of any strain solution provides the strain solution for similar discs of material with the same other properties and geometry, but any value of *k*.In the case of the stress solution, the effect of m-value involved in (31) is most significant on the distribution of the circumferential stress and the residual circumferential stress.In the case of the strain solution, the effect of m-value is most significant on the distribution of the axial strain and the residual axial strain.

The method used in the present paper is a generalization of the method developed in [31] for homogeneous discs. It is evident from the solutions provided in [31] that the method can be more successfully adopted for disc subject to other loading conditions than those used in the present paper. In particular, the basic equations derived are independent of boundary conditions. Therefore, the solution of these equations used in conjunction with any other boundary conditions (of course, the boundary value problem should be axisymmetric) supplies the distribution of stress and strain. This will be the subject of a subsequent investigation.

## Figures and Tables

**Figure 1 materials-12-00440-f001:**
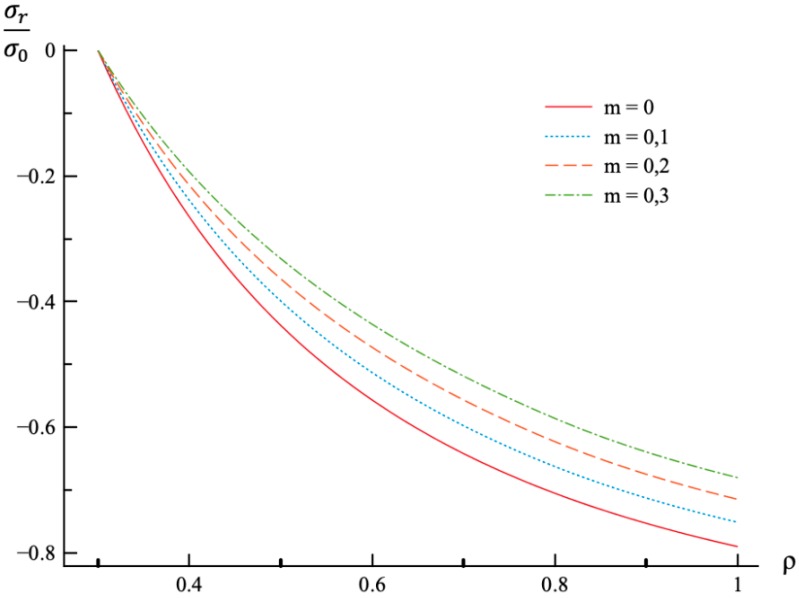
Variation of the radial stress, $\sigma_{r}$, with the dimensionless radius, ρ, at $p=p_{f}$.

**Figure 2 materials-12-00440-f002:**
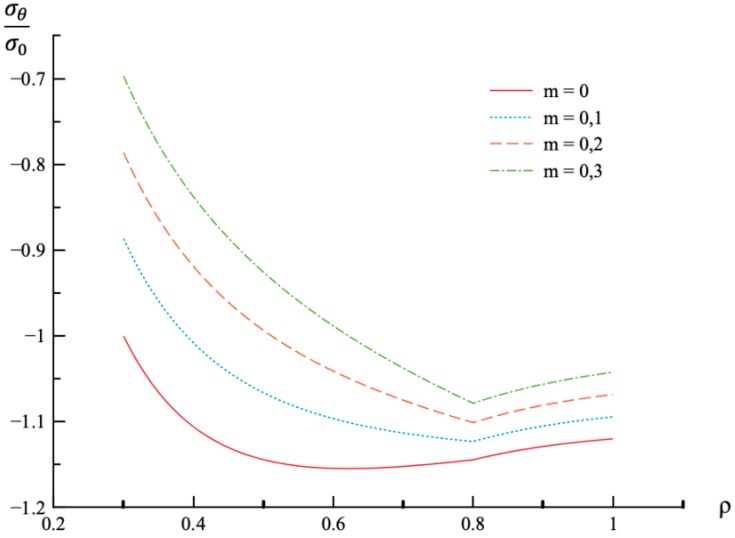
Variation of the circumferential stress, $\sigma_{\theta }$, with the dimensionless radius, ρ, at $p=p_{f}$.

**Figure 3 materials-12-00440-f003:**
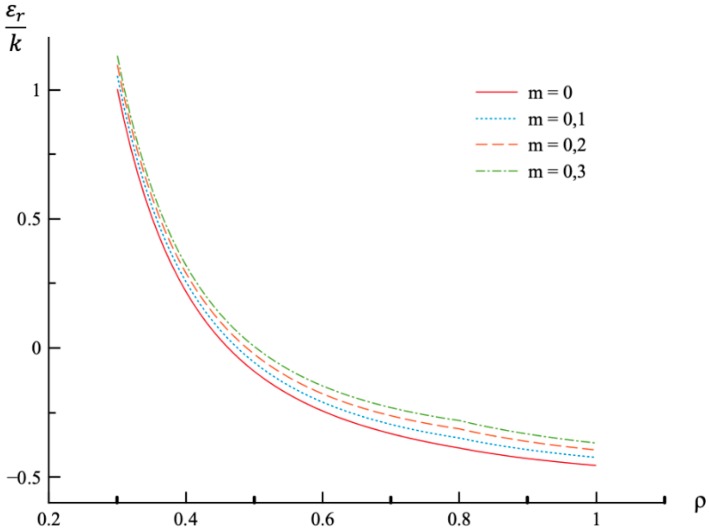
Variation of the radial strain, $\varepsilon_{r}$, with the dimensionless radius, ρ, at $p=p_{f}$.

**Figure 4 materials-12-00440-f004:**
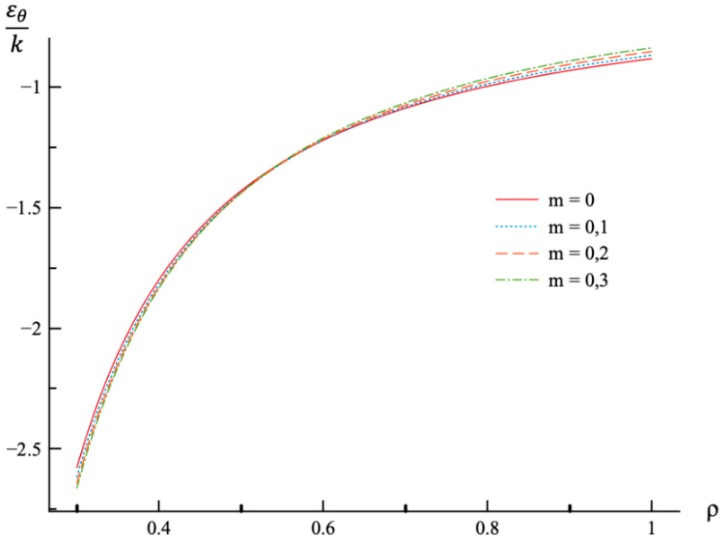
Variation of the circumferential strain, $\varepsilon_{\theta }$, with the dimensionless radius, ρ, at $p=p_{f}$.

**Figure 5 materials-12-00440-f005:**
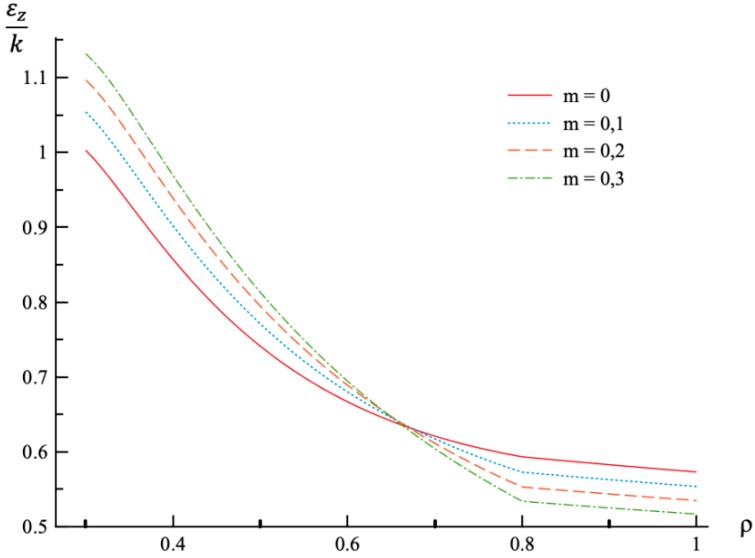
Variation of the axial strain, $\varepsilon_{z}$, with the dimensionless radius, ρ, at $p=p_{f}$.

**Figure 6 materials-12-00440-f006:**
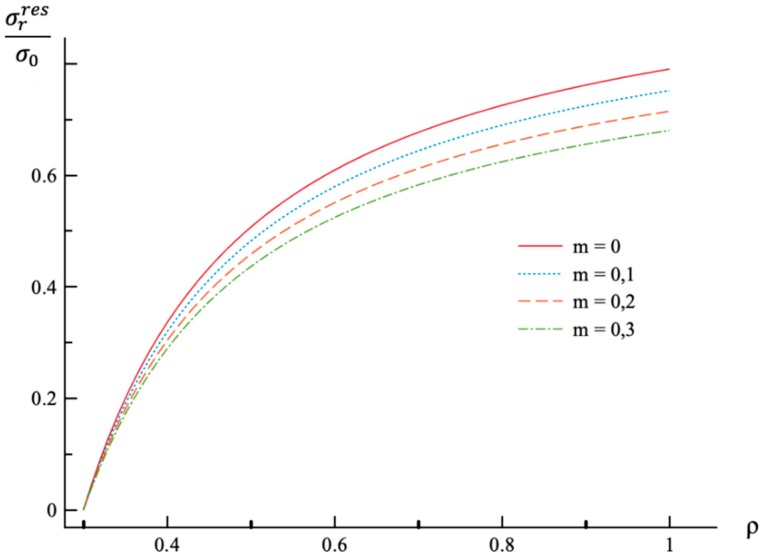
Variation of the residual radial stress, $\sigma_{r}^{res}$, with the dimensionless radius, ρ.

**Figure 7 materials-12-00440-f007:**
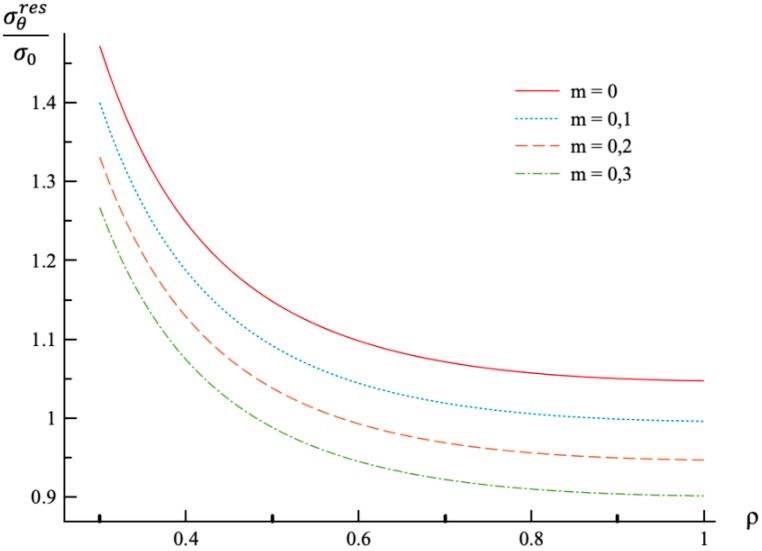
Variation of the residual circumferential stress, $\sigma_{\theta }^{res}$, with the dimensionless radius, ρ.

**Figure 8 materials-12-00440-f008:**
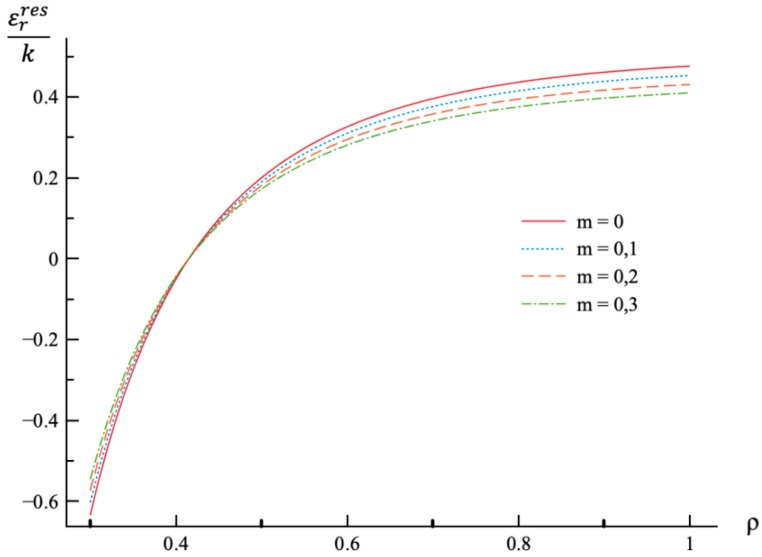
Variation of the residual radial strain, $\varepsilon_{r}^{res}$, with the dimensionless radius, ρ.

**Figure 9 materials-12-00440-f009:**
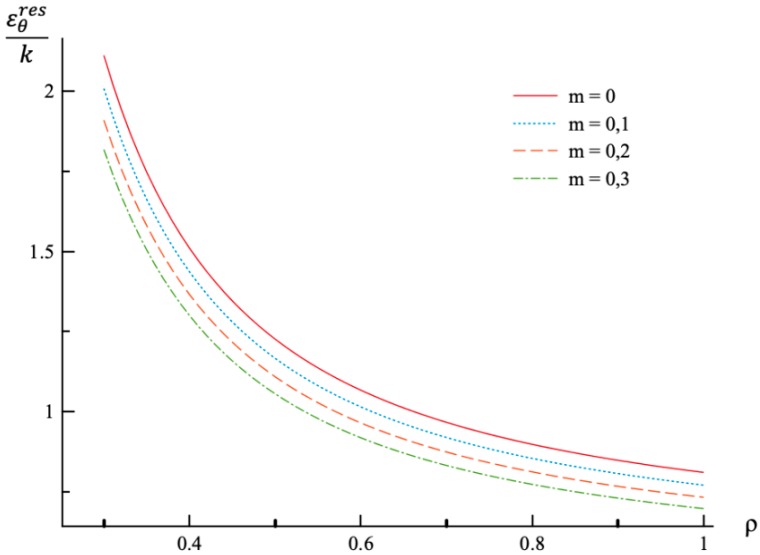
Variation of the residual circumferential strain, $\varepsilon_{\theta }^{res}$, with the dimensionless radius, ρ.

**Figure 10 materials-12-00440-f010:**
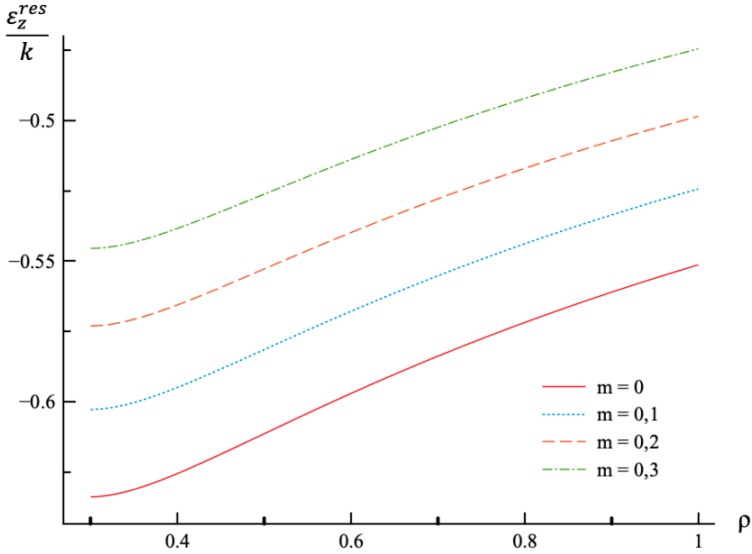
Variation of the residual axial strain, $\varepsilon_{z}^{res}$, with the dimensionless radius, ρ.

**Table 1 materials-12-00440-t001:** Dependence of the value of pressure at the end of loading on the value of m introduced in (31).

m	$p_{f}$
0	0.79
0.1	0.75
0.2	0.71
0.3	0.68

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
