# Peer review of "Description of Residual Stress and Strain Fields in FGM Hollow Disc Subject to External Pressure"

_materials, 2019, doi:10.3390/ma12030440_

Reviewer 1 Report

This is purely theoretical analysis, where the calculations are not compared with actual (experimental) results.

Author Response

All corrections made in the revised manuscript are shown in red.
Comment: This is purely theoretical analysis, where the calculations are not compared with actual (experimental) results.

Response: We agree that our paper deals with a theoretical solution. We just follow the special issue information (This Special Issue aims to collect recent studies and developments associated with mechanical problems of FGMs and FGM structures). It is evident that our paper is devoted to a mechanical problem of a FGM structure. In general, it is authors’ opinion that there is no experimental technique to verify a practically analytic solution. What is to be verified? Constitutive equations? We do not propose any new constitutive equations. We use the equations that have already been verified long time ago for different materials. Boundary conditions? We use a conventional set of boundary conditions. The remainder is pure mathematics.

Reviewer 2 Report

This manuscript solves the boundary value problem of stress distribution throughout an elasto-plastic functionally graded hollow disc with elastic properties of power-law form and a von-Mises yield criterion. The manuscript extends existing solutions to the plastic range.

Suggested Improvements: 

- 84 - grammar - in terms of

- 86 - sp - vary with radius

- 90 - not new paragraph

- 92 - under -> under a

- 94 - not new paragraph

- 97 - not new paragraph

- 104 - not new paragraph

- 114 - not new paragraph

- 124 - should -> should be

- 127 - then -> , then

- 131 - not new paragraph

- 133 - not new paragraph

- 152 - not new paragraph

- 154 - not new paragraph

- 161 - not new paragraph

- 163 - not new paragraph

- 163 - in which ... (9) -> , in which ... (9),

- 166 - not new paragraph

- 167 - has been already -> has already been

- 179 - in which ..> are valid -> , in which ... , are valid

- 184 - not new paragraph

- 190 - not new paragraph

- 194 - according -> according to

- Figures - define rho and axis variables in words in each figure - identify

  purely elastic solution in the figures

- Add a section named discussion of the results that reviews the relevance of the results with

  respect to the elastic solution - discuss both the significant and the insignificant differences -    

  much of the discussion can be moved from the present conclusions section and should be    

  divided into distinct paragraph

- Revise Conclusions - this section is too dense for a casual reader - move the detailed 

  discussion of the results to the new discussion section suggested above and only

  summarize the results in the revised conclusion section - conclusions should generally be

  brief and should not include any new statements - that is, conclusions should only repeat

  statements that have already been made in the previous sections in more concise form

- A comment on the direction of future work stemming from this study would be nice to add

Author Response

All corrections made in the revised manuscript are shown in red.
Comments and Suggestions for Authors

This manuscript solves the boundary value problem of stress distribution throughout an elasto-plastic functionally graded hollow disc with elastic properties of power-law form and a von-Mises yield criterion. The manuscript extends existing solutions to the plastic range. 

Suggested Improvements: 

- 84 - grammar - in terms of

- 86 - sp - vary with radius

- 90 - not new paragraph

- 92 - under -> under a

- 94 - not new paragraph

- 97 - not new paragraph

- 104 - not new paragraph

- 114 - not new paragraph

- 124 - should -> should be

- 127 - then -> , then

- 131 - not new paragraph

- 133 - not new paragraph

- 152 - not new paragraph

- 154 - not new paragraph

- 161 - not new paragraph

- 163 - not new paragraph

- 163 - in which ... (9) -> , in which ... (9),

- 166 - not new paragraph

- 167 - has been already -> has already been

- 179 - in which ..> are valid -> , in which ... , are valid

- 184 - not new paragraph

- 190 - not new paragraph

- 194 - according -> according to

Response: All these suggestions have been incorporated into the revised manuscript. We really appreciate the time the reviewer spent for improving our English.

- Figures - define rho and axis variables in words in each figure - identify

  purely elastic solution in the figures

Response: We have defined these variables in the captions. The solution is illustrated at p = pf > pe. The purely elastic solution does not exist at this value of p.

- Add a section named discussion of the results that reviews the relevance of the results with

  respect to the elastic solution - discuss both the significant and the insignificant differences -    

  much of the discussion can be moved from the present conclusions section and should be    

  divided into distinct paragraph

Response: The section required has been added. The general elastic solution is valid in the elastic region but it is not the purely elastic solution. The purely elastic solution does not exist if p > pe. Since the pure elastic solution is available, we do not focus on it.

- Revise Conclusions - this section is too dense for a casual reader - move the detailed 

  discussion of the results to the new discussion section suggested above and only

  summarize the results in the revised conclusion section - conclusions should generally be

  brief and should not include any new statements - that is, conclusions should only repeat

  statements that have already been made in the previous sections in more concise form

Response: The conclusions section has been shortened.

- A comment on the direction of future work stemming from this study would be nice to add

Response: The comment required has been included in the conclusions section (Conclusion 5). 

Reviewer 3 Report

This paper presents analytical studies of residual stress and strain fields in FGM hollow disc. For this purpose, flow theory of plasticity is employed.

This is no doubt an interesting paper. However, this reviewer does not recommend the publication of the manuscript in the present form because of the following reasons:

1) This reviewer think it will be useful if the authors provide some additional information on the Elastic/plastic stress solution used in this study.

2) English usage and spelling should be improved.

3) The manuscript needs more detailed description of the model.

Author Response

All corrections made in the revised manuscript are shown in red.
1)   
This reviewer think it will be useful if the authors provide some additional information on the Elastic/plastic stress solution used in this study.

Response: A new section (Section 8) has been added. In this section, a detailed description on how to calculate the distribution of stress and strain in the elastic/plastic disc is provided.

2)    English usage and spelling should be improved.

Response: We have tried our best, with great help from Reviewer #2.

3)    The manuscript needs more detailed description of the model.

Response: The constitutive equations used are summarized in the new section.

Round  2

Reviewer 1 Report

There paper has not been revised to include any experimental-calculated data comparison. As such it does not address the issue identified in round-1.

Author Response

We agree that our paper deals with a theoretical solution. We just    follow the special
  issue information (This Special Issue aims to collect recent studies    and developments
   associated with mechanical problems of FGMs and FGM structures). It    is evident that
   our paper is devoted to a mechanical problem of a FGM structure. In    general, it is the
   authors' opinion that there is no experimental technique to verify a    practically analytic
   solution.

Reviewer 3 Report

I can't see enough improvement!

Author Response

1) This reviewer think it will be useful if the authors provide some additional information on the Elastic/plastic stress solution used in this study.

Response: A new section (Section 8) has been added. In this section, a detailed description of how to calculate the distribution of stress and strain in the elastic/plastic disc is provided.

2) English usage and spelling should be improved.

Response: We have tried our best, with great help from Reviewer #2. The premium version of Grammarly has not detected any grammatical error.

3) The manuscript needs more detailed description of the model.

Response: The constitutive equations used are summarized in the new section. The reference to two textbooks on plasticity where a detailed description can be found has been added.
